# StHsfB5 Promotes Heat Resistance by Directly Regulating the Expression of *Hsp* Genes in Potato

**DOI:** 10.3390/ijms242216528

**Published:** 2023-11-20

**Authors:** Wenjiao Zhu, Chunmei Xue, Min Chen, Qing Yang

**Affiliations:** College of Life Sciences, Nanjing Agricultural University, Nanjing 210095, China; 2018116106@njau.edu.cn (C.X.); chenmincm@njau.edu.cn (M.C.)

**Keywords:** potato, StHsfB5, heat, transcription activity, *Hsp*

## Abstract

With global warming, high temperatures have become a major environmental stress that inhibits plant growth and development. Plants evolve several mechanisms to cope with heat stress accordingly. One of the important mechanisms is the Hsf (heat shock factor)–Hsp (heat shock protein) signaling pathway. Therefore, the plant transcription factor Hsf family plays important roles in response to heat stress. All Hsfs can be divided into three classes (A, B, and C). Usually, class-A Hsfs are transcriptional activators, while class-B Hsfs are transcriptional repressors. In potato, our previous work identified 27 Hsfs in the genome and analyzed HsfA3 and HsfA4C functions that promote potato heat resistance. However, the function of HsfB is still elusive. In this study, the unique B5 member StHsfB5 in potato was obtained, and its characterizations and functions were comprehensively analyzed. A quantitative real-time PCR (qRT-PCR) assay showed that *StHsfB5* was highly expressed in root, and its expression was induced by heat treatment and different kinds of phytohormones. The subcellular localization of StHsfB5 was in the nucleus, which is consistent with the characterization of transcription factors. The transgenic lines overexpressing *StHsfB5* showed higher heat resistance compared with that of the control nontransgenic lines and inhibitory lines. Experiments on the interaction between protein and DNA indicated that the StHsfB5 protein can directly bind to the promoters of target genes small *Hsps* (*sHsp17.6*, *sHsp21*, and *sHsp22.7*) and *Hsp80*, and then induce the expressions of these target genes. All these results showed that StHsfB5 may be a coactivator that promotes potato heat resistance ability by directly inducing the expression of its target genes *sHsp17.6*, *sHsp21*, *sHsp22.7*, and *Hsp80*.

## 1. Introduction

Plant growth and development are affected by many biotic and abiotic stresses in nature. With global warming due to the increased release of greenhouse gases from human industrial development, high temperatures have become a major environmental stress that affects plant morphology and physiology, eventually causing death and in turn decreasing crop yield [1,2]. The available data indicate that with every 1 °C increase in temperature, the crop yield will decrease by at least 3% [3]. Therefore, it is crucial for maintaining global food security to solve the adverse effects of heat stress.

Potato is the fourth largest food crop after rice, wheat, and corn in the world, which has a high nutritional value. For now, the global average yield of potato is far lower than its potential yield, which is mainly affected by various abiotic stresses, such as heat, cold, drought, and salt stress. As a heat-sensitive crop, potato is mainly affected in its growth, yield, and quality by heat stress. Due to global warming, this harm is even more severe [4]. The impact of high-temperature stress on potatoes is mainly through reducing the ability of tuber germination, limiting tuber induction and development, and reducing photosynthesis and the carbon element allocated to tuber growth, which cause a decrease in dry matter content and an increase in sugar alkaloids, resulting in reduced potential potato yield [4,5]. In addition to that, continuous heat stress also causes malformed potatoes (tuber cracking or internal heat necrosis) [6]. Therefore, the development of potato heat tolerance genes and the study of heat tolerance mechanisms are very important for improving the heat tolerance of potatoes, reducing the damage caused by heat stress to potato tuberization, and enabling potatoes to have a broader habitat, thereby increasing the global yield.

When temperature increases, plants respond to environmental changes by altering their own transcripts, proteins, metabolites, and lipid composition. The changes in this process are mediated by a series of signals, including Ca^2+^ signals, kinases and phosphatases, ROS (reactive oxygen species) signals, transcription factors, and hormone signals, which form a huge regulatory network, enabling plants to regain metabolic balance and achieve heat tolerance [1,7,8,9,10]. Heat shock transcription factors (Hsfs) and heat shock proteins (Hsps) play the most important roles in plant heat stress and achieving heat tolerance. Hsfs serve as terminals for signal transduction, mediating the expression of *Hsp* genes and proteins [8].

Hsfs function as transcription factors through the DNA-binding domain (DBD) in the N-terminus, binding to the heat shock response element (HSE) (5′-AGAAnnTTCT-3′) of the candidate gene promoters to regulate heat-related gene expressions and then control the plant’s response to heat stress. Many plants have identified all Hsf members in the whole genome, and there are 21 members in *Arabidopsis* [11,12], 25 in rice [11,13,14], 31 in poplar [15], 26 in soybean [16], 25 in maize [17], 56 in wheat [18], 52 in bean [19], 25 in pepper [20], 25 in tomato [21], 27 in potato [22], 22 in chickpea [23], 41 in moso bamboo [24], and 30 in common bean [25]. All these Hsfs may play an important role in plant growth and development and response to stresses.

The Hsf transcription factor family can be divided into three classes (A–C) based on the number of amino acids between HR (heptad repeats)-A and HR-B in the oligomerization domain (OD) [26]. Usually, class-A Hsfs are transcriptional activators with transcriptional activation AHA motifs (the first “A” represents aromatic amino acids W, F, and Y; “H” represents large hydrophobic amino acids L, I, V, and M; and the second “A” stands for acidic amino acids D and E), while class-B Hsfs are transcriptional repressors without AHA motifs. Most class-B Hsf members (without HsfB5s) contain the –LFGV conserved motif at the C-terminus, which is related to the function of transcriptional repression [27]. In *Arabidopsis*, HsfB1 and HsfB2b act as repressors to negatively regulate the expression of *Hsfs* like *HsfA2*, *HsfA7a*, *HsfB1*, and *HsfB2* and some *Hsp* genes that are heat-inducible; however, they positively regulate the acquired thermotolerance [28]. Meanwhile, *Arabidopsis* HsfB1 and HsfB2b may interact with HsfAs to regulate the expression of the defensin genes *PDF1.2a/b*, further controlling pathogen resistance [29]. In rice, OshsfA2c and OsHsfB4b are involved in the transcriptional regulation of *Hsp100* in the heat shock response [30]. In tomato, HsfB1, as a repressor of *HsfA1a* and *HsfA2* and a coactivator of HsfA1a, enhances protection and regulated the balance between growth and the stress response [31]. In soybean, HsfB2b improves salt tolerance through the promotion of flavonoid accumulation by activating a series of flavonoid biosynthesis-related genes and by inhibiting the repressor gene *GmNAC2* to release other genes involved in the flavonoid biosynthesis pathway [32]. These results indicate that HsfBs may function as either activators or repressors.

In potato, the research progress for its heat tolerance is slow and limited. At present, it mainly focuses on the development of potato heat-resistant genes through a homologous comparison of existing heat-resistant genes in other species [33,34] or the large-scale screening of yeast, gene chip analysis of transcriptome data after heat treatment, and QTL (quantitative trait locus) analysis of yield at high temperature [6,35,36,37]. Therefore, it is very necessary to improve the heat resistance of potatoes from a new perspective, which can expand the depth of our research on the heat resistance of potatoes. 

In our previous work, we identified all Hsfs in the potato genome [22]. These 27 Hsfs exhibit different expression patterns in response to different stresses (cold, heat, and salt). Most of the detected *Hsfs* (*Hsf001*, *004*, *005*, *007*, *008*, *009*, *014*, *018*, *024*, and *026*) were induced by heat stress [22]. Both transgenic potato lines overexpressing *Hsf005* (*HsfA3*) and *Hsf007* (*HsfA4c*) exhibited stronger heat tolerance than control [38,39]. However, the function of class-B Hsfs is still elusive.

Here, we comprehensively analyzed the characterizations of the unique B5 member StHsfB5 (also named Hsf026) in potato, and found that *StHsfB5* was highly expressed in root, and the expression was induced by heat treatment and different kinds of phytohormones. As a transcription factor, the subcellular localization of StHsfB5 was in the nucleus, which is consistent with the NLS (nuclear localization signal) domain. The transgenic lines overexpressing *StHsfB5* showed higher heat resistance compared with that of the control nontransgenic lines and inhibitory lines, and the changes of physiological indices including SOD (superoxide dismutase), POD (peroxidase) activities, MDA (malondialdehyde) content, and RWC (relative water content) were also consistent with the phenotype of the resistance. The StHsfB5 protein has a DBD domain that can bind to the target genes *sHsp17.6*, *sHsp21*, *sHsp22.7*, and *Hsp80* and promote their expression levels verified by qRT-PCR, Y1H (yeast one-hybrid) and dual-luciferase assays. All of these results show that HsfB5 as a coactivator may promote potato heat resistance ability by directly inducing the expression of its target genes *sHsp17.6*, *sHsp21*, *sHsp22.7*, and *Hsp80*.

## 2. Results

### 2.1. Cloning and Sequence Analysis of StHsfB5

The coding sequence (CDS) of *StHsfB5* was cloned from potato cultivar Désirée by PCR, and the results showed that the length of *StHsfB5* ranges from 500 to 750 bp, which is consistent with its predicted length of 606 bp (Figure 1A). Then, the sequencing results showed that there are differences between the reference sequence and the actual sequence in four bases which are ^61^T–A, ^95^T–C, ^248^T–C, and ^516^T–A (Figure 1B). Three changes, ^61^T–A, ^95^T–C, and ^516^T–A, caused the changes of amino acids (Figure 1C). 

StHsfB5 proteins contained the conserved DBD and NLS domains, and there was no amino acid between HR-A and HR-B, which is consistent with the characterization of Hsf class-B (Figure 2A). However, there is no -LFGV conserved sequence in the C terminus of this HsfB5 protein. The alignment of HsfB5 showed that the sequence of StHsfB5 was similar to that in tomato (*Solanum lycopersicum*) (Figure 2A). The phylogenetic trees showed that HsfB5 in potato is closely related to that in the same *Solanaceae family*, like tomato, tobacco (*Nicotiana tabacum*), and pepper (*Capsicum annuum*) (Figure 2B). 

### 2.2. The Predicted Structure of StHsfB5 Protein

The online analysis software NPS@ was used to analyze the secondary structure (Figure 3A), and the results show that the α-helix accounts for more than 50% and is mainly concentrated in the C-terminal, while the frequency of β-sheet is small and random coil is 38.81%.

The tertiary structure of StHsfB5 was predicted by Swiss model using the homologous modeling method with the template—DNA-binding protein (AlphaFold DB model of M1CWQ4_SOLTU (potato gene: 102595317)) (Figure 3B), the results were consistent with the secondary structure prediction, the N-terminal of the protein was mainly random curl, and the C-terminal was mainly α-helix, which was suitable for the function of the StHsfB5 protein. This predicted structure is similar to that of SlHsfB5 in tomato using the different template—HSF-type DNA-binding domain-containing protein (AlphaFold DB model of A0A6N2AL34_SOLCI (tomato gene: A0A6N2AL34_SOLCI).

### 2.3. Tissue Expression and the Expression Changes of StHsfB5 under Heat Stress and Different Hormones

To analyze the expression of *StHsfB5* in different tissues, including root, stem, leaves, stolon, and tuber, the qRT-PCR assay was deduced and the results showed that the expression level of *StHsfB5* is the highest in roots, and is more than that in stem, leaf, tuber, and stolon (Figure 4A), which indicates that StHsfB5 mainly functions in the root. In addition, the expression of *StHsfB5* was significantly induced in the high temperature with time, and reached the peak at 24 h (Figure 4B), which indicates that StHsfB5 plays a positive role in resistance to heat stress. Moreover, the expression of *StHsfB5* was induced by all five tested hormones, including IAA (3-indoleacetic acid, one kind of auxin), SA (salicylic acid), GA (gibberellins), ABA (abscisic acid), and BR (brassinosteroid) (Figure 4C), and this result implies that StHsfB5 is important in hormone signal pathways.

### 2.4. The Promoter Analysis of StHsfB5

The expression of *StHsfB5* gene was induced by different hormones and heat stress, which are regulated on the transcript levels. Therefore, we analyzed the promoter sequence of *StHsfB5*, and the results showed that there are many functional elements, including hormones ABA, GA, BR, and MeJA (methyl jasmonate)-related elements, stress elements in drought and anaerobism, and other elements involved in light response and circadian rhythm control (Table 1). This result shows that StHsfB5 not only plays an important role in stresses and hormone signal pathways, but also functions in environmental regulation. 

### 2.5. The Subcellular Localization of StHsfB5

As we know, the transcription factor functions in the nucleus, so we analyzed the subcellular localization of StHsfB5 (Figure 5) and the results showed that StHsfB5 localized in the nucleus, which is consistent with the predict results by the online software Plant-mPLoc (http://www.csbio.sjtu.edu.cn/bioinf/plant-multi/, accessed on 5 May 2021). This may imply that StHsfB5 can function as a transcription factor in regulating the plant growth and development and response to stresses.

### 2.6. Construction of Transgenic Plants Overexpressing or Inhibiting Expression of StHsfB5

To further analyze the effects of StHsfB5 on potato, we constructed the transgenic plants by *Agrobacterium* transformation method and identified the positive lines by PCR (Figure 6). We inserted the CDS (coding sequence) sequence or CDS-SRDX (a chimeric repressor, adding a peptide tail LDLDLELRLGFA at the C-terminal of proteins) [27,40] of *StHsfB5* into the pCAMBIA1305 vector (Figure 6A). Through the transgenic method, we obtained 12 overexpressing lines and 2 inhibiting expressing lines, and PCR analysis showed that there are corresponding fragments inserted in the genome of potato overexpressing lines OE3~9 and OE12 (Figure 6B), and inhibiting lines SRDX-1 and SRDX-2 (Figure 6C). Further, qRT-PCR assay verified that the selected six OE lines and two SRDX lines expressed higher levels than the control (Figure 6D). We chose three overexpression lines, OE-3, OE-4, OE-5, and an inhibiting-expression line, SRDX-1, as a negative control for further analysis due to their high expression levels.

### 2.7. StHsfB5 Overexpression Increases Potato Resistance to Heat Stress

To verify StHsfB5 functions in response to heat stress, the one-month-old plantlets were cultivated in water at 45 °C/24 °C for three days after cultivation for 7 days at normal temperature, and the results showed that there was a significant change in phenotype of nontransgenic lines under heat stress, specifically the stem segment near the top, which has the water loss wilting leading the terminal bud drooping, while in overexpression lines OE-3 and OE-5, there were no obvious changes such as in nontransgenic lines (Figure 7A,B). SRDX-1 showed a similar phenotype with the nontransgenic lines, which is consistent with the fact that StHsfB5-SRDX can competitively bind to the HSE elements, which inhibits the StHsfB5 functions (Figure 7B). The OE-4 lines also showed a similar phenotype to the control, which may be due to the wrong insertion of the gene in the genome. Next, we tested a series of physiological indices for stress, including SOD activity (Figure 7C), POD activity (Figure 7D), MDA content (Figure 7E), and the RWC (Figure 7F), because there is a positive correlation among heat tolerance and SOD activities, POD activities, and RWC contents, while there is a negative correlation among heat tolerance and MDA contents, which can show the different heat tolerance [38,39,41]. The results showed that the SOD and POD activities and RWC contents were higher in overexpression lines than control and SRDX-1 lines, while MDA contents were lower in overexpression lines than control and SRDX-1 lines (Figure 7C–F). All these results indicate that StHsfB5 overexpression can promote potato resistance to heat stress.

### 2.8. StHsfB5 Regulates the Expression of Candidate Target Genes

As StHsfB5 has a DNA-binding domain, we analyzed the HSE domain in the candidate target genes, including *sHsp17.6*, *sHsp21*, *sHsp22.7*, *sHsp23.6*, and *Hsp80*, and the results showed that in the promoters of these genes, there exists classical HSE elements (Figure 8A), which should be further studied. First, we tested the expression levels in the transgenic lines, and the results showed that overexpressing *StHsfB5* can induce the expression levels of *sHsp17.6*, *sHsp21*, *sHsp22.7*, and *Hsp80*, while heat treatment induced the higher levels of *sHsp17.6*, *sHsp21*, and *sHsp22.7* (Figure 8B), which implies that *sHsp17.6*, *sHsp21*, *sHsp22.7*, and *Hsp80* may be the candidate genes of StHsfB5. 

Next, we used both Y1H and dual-luciferase assays to validate that HsfB5 can bind to the promoters of these four candidate genes (Figure 9), which verified that HsfB5 improves the heat resistance by directly promoting the expression levels of *sHsp17.6*, *sHsp21*, *sHsp22.7*, and *Hsp80*.

## 3. Discussion

In our studies, we verified the proper sequence of StHsfB5 from potato cultivar Désirée, by multiple PCRs which collect three samples, using three high-fidelity DNA polymerases under increasing annealing temperature. For these attempts, all sequencing results were the same, and there were four bases different from the reference sequences from the SPUD DB website. This may be due to the fact that potatoes, as tetraploids, exhibit polymorphism in gene loci, and the genome reference sequences were collected from the doubled monoploid potato *S. tuberosum* Group Phureja DM 1–3 516 R44, not from Désirée.

As a class-B member, StHsfB5 is a special class that has no tetrapeptide -LFGV- in the C-terminal. Therefore, StHsfB5 was separated from the B2 class with the -LFGV- domain, which is assumed to function as a repressor by interaction with another corepressor in the transcription machinery [12]. Hsf B5 members were found in many plants; however, the numbers were low, like one in potato (PGSC0003DMG400029718), two in soybean (Glyma13g24860 and Glyma01g34490), one in tomato (Solyc02g078340), two in poplar (Potri.004G042600 and Potri.011G051600), and one in castor bean (Rc29851.t000049). The protein HsfB5 does not even exist in some plants, like *Arabidopsis*, rice, and maize. These results imply that HsfB5 may have a unique function in plant growth and development.

In *Arabidopsis*, overexpressing *HsfB1* plants had tiny rosettes with small, crinkly leaves and crooked roots compared to the wild-type plants [28], which showed that HsfB1 is important for root development. In our studies, the expression of *StHsfB5* was also higher in the root, and another potato HsfA member, HsfA4c, had the highest expression levels in adult leaves [39], and overexpressing *HsfB5* and *HsfA4c* both showed enhanced heat resistance, which indicated that when potatoes encounter the heat stress, they may arouse different signaling pathways in different organs to synergistically resist this stress to ensure they can survive. 

Whether StHsfB5 has transcriptional activity is worth exploring due to its specificity. The results of our validation experiment showed that StHsfB5 has no transcription activities in Y1H assay, while StHsfB5 has no transcription repression abilities in dual-luciferase assay (not shown). This result appears to be contradictory, which may be due to the different environments for protein expression. Further experiments showed that HsfB5 can promote potato heat resistance and can bind to the target genes *sHsp17.6*, *sHsp21*, *sHsp22.7*, and *Hsp80* to induce their expression, which confirms the transcriptional activity of StHsfB5. In addition, Hsfs function by trimerization to bind to heat shock elements present in the promoter region of target genes [42], which implies that StHsfB5 may function as a coactivator by forming a heterotrimer with other Hsf A members to regulate the expression of *Hsp*s, further promoting the heat resistance of potato. Further experiments should be designed to seek the Hsf A members, which can interact with HsfB5.

With the development of global warming, heat stress is becoming a vital environmental factor that affects the crop production and quality. In potato, heat stress severely reduces the mass of the tubers [5]. Therefore, the function analysis of StHsfB5 supplemented and improved the potato heat tolerance mechanism, and after excluding the safety issues of genetically modified lines and the possible negative role impact of StHsfB5 on potato tuber yield, the overexpressing StHsfB5 lines would be a potential heat-tolerance material that are suitable for broader habitats like Africa or for more growth cycles in summer to increase production to solve the global food crisis.

## 4. Materials and Methods 

### 4.1. Preparation of Plant Materials and Growth Conditions 

Potato cultivar Désirée (*Solanum tuberosum* L.) and tobacco (*Nicotiana benthamiana*) were preserved in our laboratory. The potato plantlets were cultured in MS medium [43] containing 3% sucrose and 0.8% agar at pH 5.8. They were grown in a greenhouse with a day/night regime of 16/8 h at 24/20 °C. Subculture was conducted every 4 weeks. 

The one-month-old leaves were collected for extraction of total RNA and genomic DNA. The detached leaves from one-month plantlets were incubated with different hormones with different concentrations (0.5 μM IAA, SA, GA3, ABA or 0.3 μM epi-BR) for 2 h. The root, stem, leaves, stolon, and tuber were collected from the two-month plants for spatial expression of *StHsfB5*. The one-month plantlets were cultured with half-strength Hoagland solution for another 1 week before being treated with heat stress (35 °C for 0 h, 2 h, 6 h, 12 h, and 24 h). All these treated samples were collected immediately, immersed in liquid nitrogen, and stored at −80 °C prior to RNA extraction.

### 4.2. RNA Extraction and Gene Expression Pattern Analysis of StHsfB5

Total RNA was extracted from the tissues by TRIzol reagent (Invitrogen, Waltham, MA, USA). Briefly, 0.5 μg integrated and high-quality RNA was reversely transcribed to the first-strand cDNA using the PrimeScript™ RT reagent Kit with gDNA Eraser (Perfect Real Time) (Takara, Kusatsu, Japan).

For quantitative real-time PCR, the reaction was carried out with the TB Green^®^ Premix Ex Taq™ II (Tli RNaseH Plus) (Takara) on an ABI7500 Real Time PCR System (Applied Biosystems, Waltham, MA, USA). According to the manual, the reaction volume was 20 μL with the specific primers and the reference primers of the gene *EF1α* (Appendix A) following the reaction program of 95 °C for 1 min, then 40 cycles of 95 °C for 15 s, and 60 °C for 30 s. The expression levels were normalized using the 2^−ΔΔCt^ method. For each sample, three experimental replicates and three biological repeats were performed. The method of Dunnett’s two-tailed *t*-test was used for the statistical analysis of qRT-PCR results, and the significant differences were shown as * (*p* < 0.05) or ** (*p* < 0.01).

### 4.3. Gene Clone and Vector Constructions

cDNA was reversely transcribed from the extracted total RNA using a BioTeke supermoIII RT Kit (BioTeke, Beijing, China), which was used for gene cloning. *StHsfB5* and *StHsfB5-SRDX* genes were amplified using specific primers (Appendix A) by Phanta Max Super-Fidelity DNA Polymerase (Vazyme, Nanjing, China). The PCR program was 95 °C for 3 min, 35 cycles of 95 °C 15 s, 58 °C for 15 s, 72 °C for 1 min, and 72 °C for 5 min. Subsequently, the product of PCR was gel-purified with an AxyPrep Gel DNA Extraction Kit (Axygen, Hangzhou, China) and inserted into the pCAMBIA1305 vector (between Spe I and BamH I sites).

The total genomic DNA was extracted using the new rapid plant genome DNA extraction kit (BioTeke). The 1.5 kb upstream sequences from the start codon of each *Hsp* (*sHsp17.6*, *sHsp21*, *sHsp22.7* and *Hsp80*) gene as a promoter were cloned and then inserted into the pGreenII 0800 vector (between Kpn I and Sal I sites).

For Y1H, StHsfB5 was cloned and then inserted into the pGADT7 vector (between Sfi I and Sma I sites), while the 40–50 bp sequences containing at least one HSE motif in the *Hsp* promoters were synthesized and inserted into the pABAi vector (between Hind III and Kpn I sites). All these vectors were verified by sequencing.

### 4.4. Protein Structure Analysis of StHsfB5

The secondary structure was predicted at the NPS@ website (http://npsa-prabi.ibcp.fr/cgi-bin/secpred_sopma.pl, accessed on 21 May 2021), and the tertiary structure was analyzed at the SWISS-MODEL website (http://swissmodel.expasy.org, accessed on 9 November 2023).

### 4.5. Alignment and Phylogenetic Tree Construction of HsfB5 Proteins

HsfB5 members of tomato (*Solanum lycopersicum*), soybean (*Glycine max*), grapevine (*Vitis vinifera*), tobacco (*Nicotiana tabacum*), pepper (*Capsicum annuum*), soybean (*Glycine max*), poplar (*Populus trichocarpa*), and castor bean (*Ricinus communis*) were collected from a platform for heat stress transcription factors (HSFs)—HEATSTER (https://applbio.biologie.uni-frankfurt.de/hsf/heatster/HSFS.php?DB=1#, accessed on 6 June 2023). The alignments of ten HsfB5 amino acid sequences were deduced by DNAMAN software, and the phylogenetic tree was deduced by ClustalX2 and Mega 7.0 software with 1000 bootstraps using NJ methods.

### 4.6. Subcellular Localization of StHsfB5 Protein

For transient expression, the constructed plasmid StHsfB5-pCAMBIA1305 with GFP tag and the control empty vector pCAMBIA1305 were transformed into *Agrobacterium tumefaciens* GV3101, and then were suspended with buffers to OD600 = 0.2. The resuspension buffers carrying StHsfB5-GFP or empty vector were injected into 4-week-old *Nicotiana benthamiana* leaves. After infiltration for 3 days, including 1 day in the dark and 2 days in the normal rhythm, the lower-leaf epidermis cells were used for observing the green fluorescence at 488 nm by confocal microscopy (Zeiss, Jena, Germany).

### 4.7. Transformation and Identification of Transgenic Potato

The constructed StHsfB5-pCAMBIA1305 and StHsfB5SRDX-pCAMBIA1305 were transformed into *Agrobacterium tumefaciens* GV3101. Désirée transformation was performed using the following steps: 1 cm stem segments with one leaf were collected as explants, then infection, callus and shoot induction, rooting, acclimation of plantlets, and growth in pots was performed. Identification of transgenic plants was performed by PCR using the designed primers containing both the partial pCAMBIA1305 and StHsfB5 CDS (Appendix A). The PCR program was 95 °C for 3 min, 35 cycles of 95 °C 15 s, 58 °C for 15 s, 72 °C for 1 min, and 72 °C for 5 min. Further qRT-PCR assay was deduced to detect the expression levels of StHsfB5 or *StHsfB5-SRDX* in the positive lines. The calculation of the related expression level was described as above.

### 4.8. Measurement of Physiological and Biochemical Indices Related to Heat Resistance of Potato Plantlets

The heat tolerance of *StHsfB5* overexpression lines, StHsfB5-SRDX overexpression lines, and nontransgenic lines of Désirée were evaluated using the one-month-old plants following one week of growth in the half-strength Hoagland solution. The heat treatment conditions were day 16 h (light intensity 100% or 22000LX) at 45 °C; dark 8 h (light intensity 0) at 24 °C. The total time for heat treatment was 3 days. Photos were taken of the plants after 3 days of heat treatment. Then, their leaves were taken for testing superoxide dismutase (SOD) and peroxidase (POD) activity, malondialdehyde (MDA) content, and relative water content (RWC). We ground 0.2 g samples in a precooled mortar, and added 1.6 mL precooled PBS (pH7.8). Then, we mixed and centrifuged at 4 °C and 12,000 rpm for 20 min, and collected the supernatant for the following tests. The measurements of SOD activity and MDA content were the same as those in reference [39]. The SOD activity was analyzed using the nitrogen blue tetrazole photochemical reduction method, and the activity unit of SOD equaled the amount of enzyme that can inhibit the 50% photochemical reduction of nitrogen blue tetrazole. The formula for calculating SOD activity is as follows: SOD activity = [(A_CK_ − A_E_) × V]/(1/2A_CK_ × W × Vt), where A_ck_: the absorbance of control; A_E_: the absorbance of samples; V: the volume of total samples; Vt: the volume of the tested samples; W: the fresh weight of all samples. The steps for testing RWC were as follows: We immersed 0.5 g (fresh weight, FW) potato fresh leaves into water for a period of time, took them out, absorbed the surface moisture with filter paper, and weighed them; then immersed the leaves into water for a period of time, took them out again, absorbed the surface moisture with filter paper, and weighed them again; then repeated the operation several times until the weight stabilized, at which point the weight of the leaves was the saturated fresh weight (SFW). Afterwards, we placed the saturated leaves into a paper bag and placed them in an oven at 105 °C for 10 min to blanch. We adjusted the oven temperature to 75 °C and continued drying the leaves until their weight no longer changed. At this point, we removed the blade, cooled the leaves to room temperature, and recorded their dry weight (DW). The formula for calculating RWC is as follows: RWC = (FW − DW)/(SFW − DW) × 100%. All these experiments were repeated three times.

### 4.9. Y1H Assay for Interaction between StHsfB5 Protein and the Promoter of Hsp Genes

Hsp (bait)-pABAi plasmids were transformed into yeast strain Y1H Gold cells, respectively, and were cultured at the SD −Ura at 30 °C for 3–4 days, then the positive cells were screened by colony PCR identification. Then, we transformed the recombinant plasmid pGADT7-HsfB5 into Y1H Gold competent cells that contained one Hsp-pABAi plasmid, and these cells were cultured at the SD −Leu at 30 °C for 3–4 days. The colony was identified with PCR, and was selected for the positive cells. Two different cells containing the only Hsp-pABAi plasmid or two plasmids (pGADT7-HsfB5 and Hsp-pABAi) were cultivated to OD600 = 0.6, then diluted separately by 1×, 10×, 100×, and 1000× and were grown on the SD −Ura or SD −Leu with 0, 100, 200, 300, 400, 500, and 600 ng·mL^−1^ AbA at 30 °C for 3–5 days separately.

### 4.10. Dual-Luciferase Assay for StHsfB5 Protein and the Promoter of Hsp Genes

Four proHsp-pGreenII0800 plasmids were transformed into *Agrobacterium tumefaciens* GV3101 (p19-pSoup), and then were suspended with buffers (MES 10 mM, MgCl_2_ 10 mM, AS 0.2 mM) to OD600 = 0.5~1. The GV3101 cells containing StHsfB5-pCAMBIA1305 or pCAMBIA1305 plasmid were also suspended like the above. We mixed the bacterial solution according to the volume ration of promoter-0800 and proteins StHsfB5 or control = 1:7–1:9, and activated it in the dark for about 3 h at 28 °C. Then, we infected the tobacco with these mixtures, and cultivated for 2–3 days normally for observation. We sprayed D-fluorescein potassium salt (Beyotime, Shanghai, China) with 200 μg/mL concentration directly on the surface of the lower leaves, and incubated them in the dark for about half an hour, then tested the fluorescence intensity using the Tanon 5200 Series Fully Automatic Chemiluminescence Image Analysis System.

## 5. Conclusions

All our results indicated that StHsfB5 functions in promoting potato heat resistance by inducing the expression of *Hsp*s, which not only reveals the mechanism of StHsfB5 regulating potato heat resistance, but also provides a candidate heat-resistant gene for improving potato varieties.

## Figures and Tables

**Figure 1 ijms-24-16528-f001:**
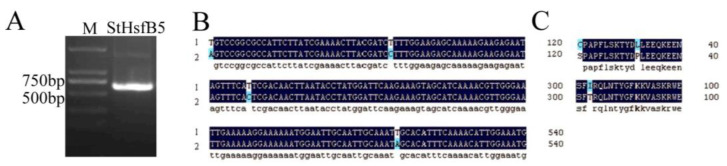
The DNA sequence analysis of *StHsfB5*. (**A**) Agarose gel electrophoresis of *StHsfB5* PCR amplification product. M: 2000bp DNA marker. (**B**) Partial DNA sequencing alignment analysis of *StHsfB5*. (**C**) Partial protein alignment analysis for translated StHsfB5 protein.

**Figure 2 ijms-24-16528-f002:**
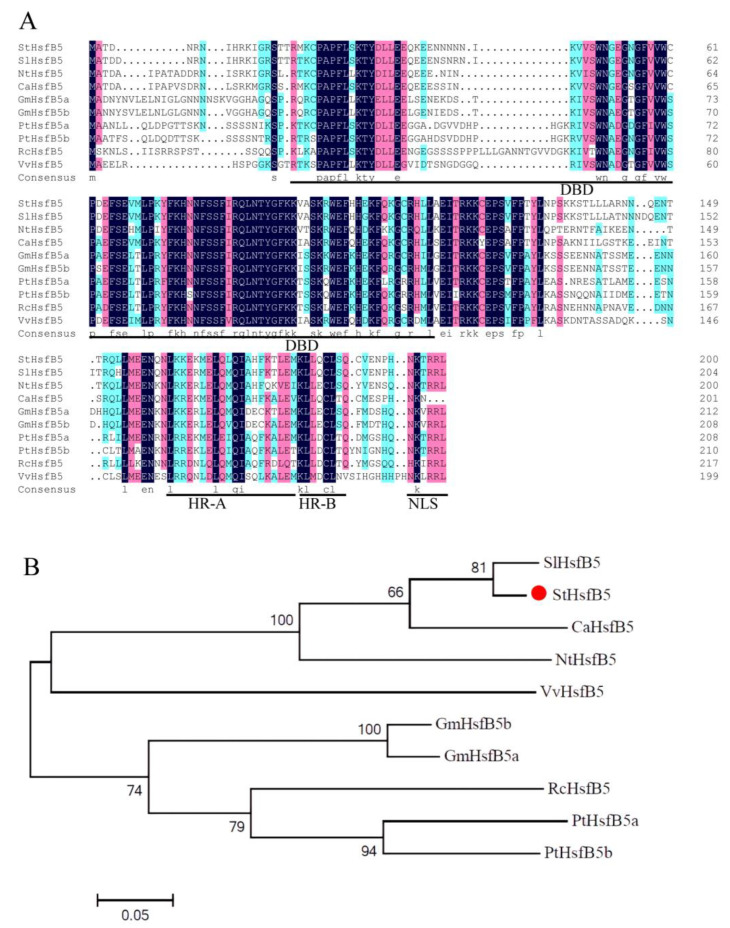
Protein sequence alignment and phylogenetic analysis of StHsfB5. (**A**) Sequence alignment analysis of HsfB5s from different species. StHsfB5 (*Solanum tubersosum*): PGSC0003DMG400029718; SlHsfB5 (*Solanum lycopersicum*): Solyc02g078340; VvHsfB5 (*Vitis vinifera*): VV78X139014.10, NtHsfB5 (*Nicotiana tabacum*): XP_009629680.1; CaHsfB5 (*Capsicum annuum*): XP_016561996.1; GmHsfB5a (*Glycine max*): Glyma13g24860; GmHsfB5b (*Glycine max*): Glyma01g34490; PtHsfB5a (*Populus trichocarpa*): Potri.004G042600; PtHsfB5b (*Populus trichocarpa*): Potri.011G051600; RcHsfB5 (*Ricinus communis*): Rc29851.t000049. (**B**) Phylogenetic trees of these 10 HsfB5 proteins. The red circle highlights StHsf5.

**Figure 3 ijms-24-16528-f003:**
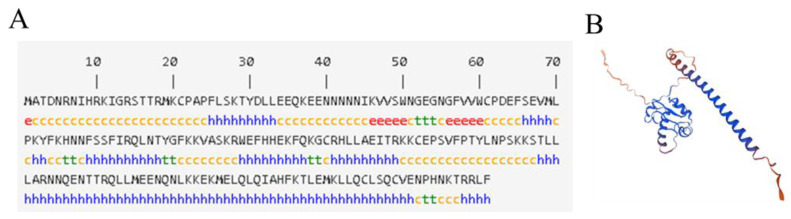
The advanced structure of proteins of StHsfB5. (**A**) Secondary structure of StHsfB5 protein. h: α-helix (blue); e: β-sheet (red); c: random coil (yellow). t: hydrogen bonded turn (green). (**B**) Tertiary structure of StHsfB5 protein.

**Figure 4 ijms-24-16528-f004:**
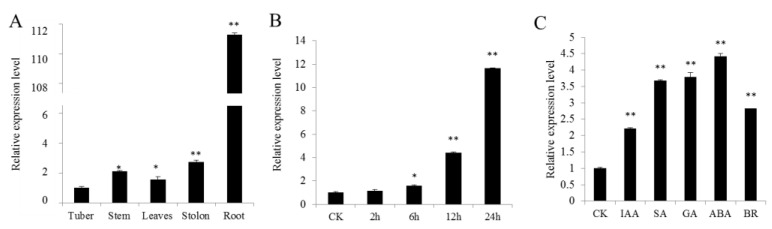
Analysis of the expression pattern of StHsfB5. (**A**) Expression pattern of StHsfB5 in different tissues, including tuber, stem, leaves, stolon, and root. (**B**) StHsfB5 expression in leaves after heat treatment at 35 °C for different times (0, 2, 6, 12, 24 h). (**C**) StHsfB5 expression in detached leaves under different hormones treatment for 2 h. CK means control, * means *p* < 0.05, and ** means *p* < 0.01.

**Figure 5 ijms-24-16528-f005:**
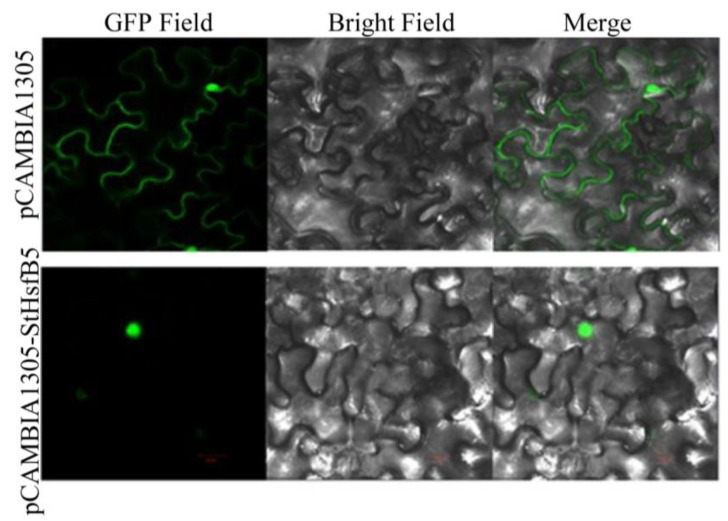
Subcellular localization of StHsfB5. The empty vector pCAMBIA1305 was used as control. The green dots represent fluorescence of the expressed StHsfB5-GFP or GFP.

**Figure 6 ijms-24-16528-f006:**
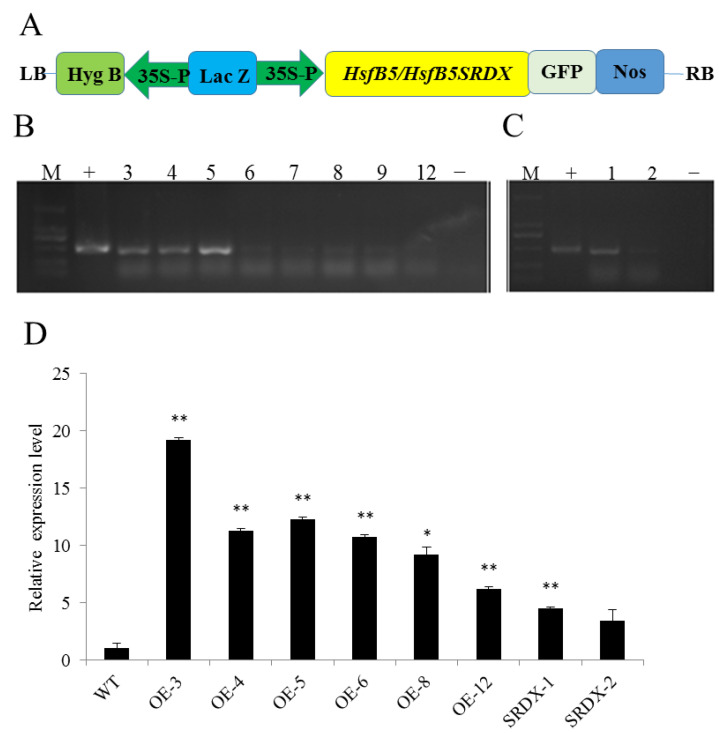
Construction and identification of StHsfB5/StHsfB5SRDX-pCAMBIA1305 transgenic lines. (**A**) Schematic diagram of StHsfB5/StHsfB5SRDX-pCAMBIA1305 expression vectors. (**B**) Identification of StHsfB5-pCAMBIA1305 transgenic potato plants by PCR. M: DL2000; +: plasmid positive control; −: wild-type potato plants; numbers 3–9, 12: overexpression lines. (**C**) Identification of StHsfB5SRDX-pCAMBIA1305 transgenic potato plants by PCR. M: DL2000; +: plasmid positive control; −: wild-type potato plants; numbers 1–2: inhibitory expression lines. (**D**) Expression levels of transgenic lines by qRT-PCR. WT: Désirée; OE3~12: StHsfB5 overexpression lines. SRDX-1~2: inhibitory expression lines. Bars show standard deviation. *: *p* < 0.05; **: *p* < 0.01.

**Figure 7 ijms-24-16528-f007:**
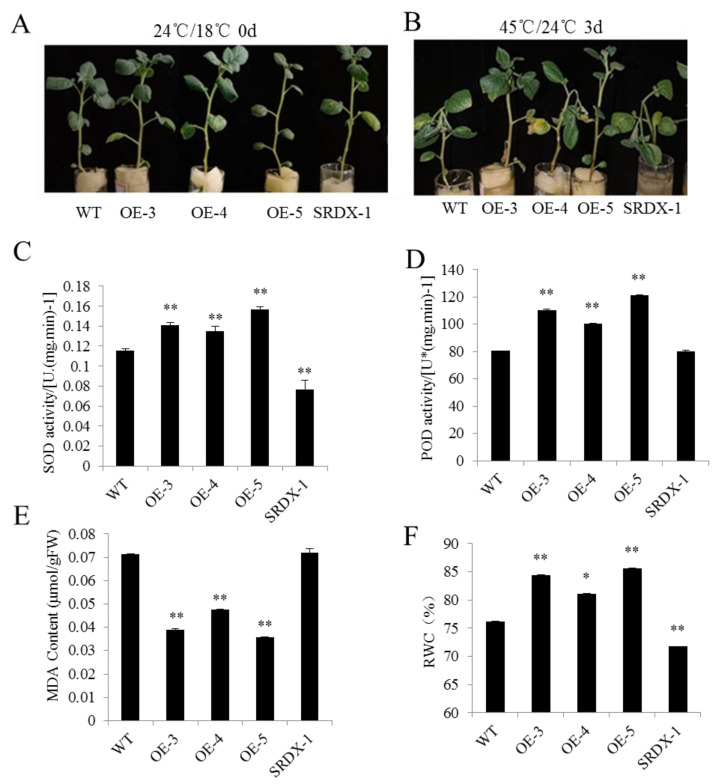
Analysis on the heat tolerance of transgenic lines. (**A**) The phenotype of potato lines before heat treatment. (**B**) The phenotype of potato lines after heat treatment (45 °C/24 °C for 3 days). (**C**–**F**) Analysis on heat-related physiological and biochemical indices of transgenic lines. (**C**) SOD activity. (**D**) POD activity. (**E**) MDA content. (**F**) RWC. Bars show standard deviation. *: *p* < 0.05; **: *p* < 0.01.

**Figure 8 ijms-24-16528-f008:**
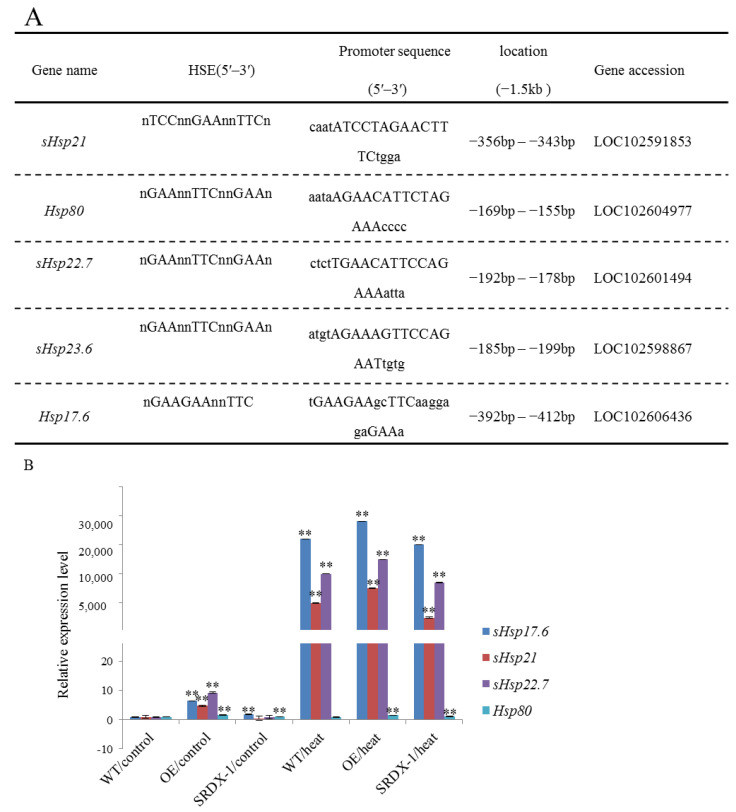
*Hsps* are the candidate target genes of StHsfB5. (**A**) Analysis on the HSE motifs in the promoter of *Hsps*. (**B**) The expression changes of *Hsps* in the transgenic lines compared with WT at normal temperature or heat conditions by qRT-PCR. Bars show standard deviation. **: *p* < 0.01.

**Figure 9 ijms-24-16528-f009:**
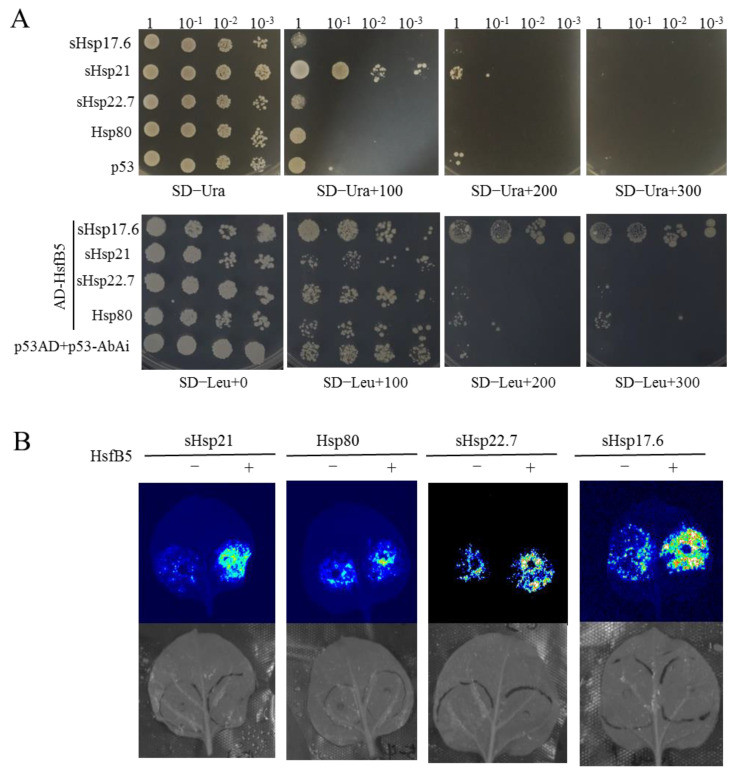
StHsfB5 can bind to the promoters of *Hsps*. (**A**) Yeast one-hybrid assays for StHsfB5-BD binding to the HSE motifs in the promoters of *Hsps*. (**B**) Dual-luciferase assays for HsfB5 inducing the fluorescent expression of proHsp-0800s.

**Table 1 ijms-24-16528-t001:** Functional element analysis of *StHsfB5* promoter region.

No.	Elements	Number	Function Prediction
1	ABRE (ACGTG)	5	ABA response
2	P-box (CCTTTTG)	1	GA response
3	E-BOX (CANNTG)	8	
	BRRE (CGTGYG)	1	BR response
4	MBS (CAACTG)	1	Drought induction
5	TC-rich repeats (ATTCTCTAAC/GTTTTCTTAC)	2	Stress induction
6	ARE (AAACCA)	4	Anaerobic induction
7	CGTCA-motif (CGTCA)	1	MeJA response
	TGACG-motif (TGACG)	1	
8	TCT-motif (TCTTAC)	2	Light response
9	AE-box (AGAAACAA/AGAAACTT)	2	Light response
	Box 4 (ATTAAT)	4	
	G-Box (CACGTT/TACGTG/TAACACGTAG/CACGAC)	7	
10	Circadian (CAAAGATATC)	1	Circadian rhythm control

## Data Availability

The original datasets described in the study are included in the article/Appendix A. Further inquiries can be addressed to the corresponding authors.

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
