# Peer review of "StHsfB5 Promotes Heat Resistance by Directly Regulating the Expression of Hsp Genes in Potato"

_ijms, 2023, doi:10.3390/ijms242216528_

Round 1
Reviewer 1 Report
Comments and Suggestions for Authors
In present MS, the authors have described the role of heat shock factors (Hsfs), specifically HsfB, in response to heat stress in potato plants. It mentions the classification of Hsfs into three classes (A, B, C), with Class A being activators and Class B being repressors. The study focuses on a specific member, StHsfB5, and analyzes its expression, subcellular localization, and its impact on heat resistance. The results suggest that StHsfB5 functions as a co-activator that promotes heat resistance by directly regulating the expression of certain target genes.
Critique:
- Author need to provide context of the study about the broader field of plant biology or the significance of heat stress response in abstract and introduction. Some background information would help readers understand the relevance of the study.
- The text uses numerous abbreviations without prior explanation. This may be confusing to readers unfamiliar with these terms. Providing a glossary or explaining abbreviations upon first use is essential.
- Lack of Specifics: While the text mentions results like the higher heat resistance in transgenic lines overexpressing StHsfB5, it lacks specific data or statistical information to support these claims. The absence of quantitative results makes it difficult to evaluate the robustness of the findings. Wherever possible authors need to provide data; refer to figures and tables appropriately or cite the published scientific literature.
- The text repeats certain phrases, such as "promote potato heat resistance" and "promoting the expression of its target genes." This repetition can be reduced to improve the overall readability.
- The text concludes that StHsfB5 may be a co-activator that promotes heat resistance, but it would be helpful to discuss the broader implications of this finding and its potential significance in the context of plant biology and crop improvement.
The writing style is somewhat convoluted and lacks clarity. It would benefit from simplifying sentence structures and improving the flow of information. For example, the sentence "Both Y2H assay and dual-luciferase assay indicated that HsfB5 protein can directly bind to the target genes sHsp17.6, sHsp21, sHsp22.7 and Hsp80 and promote their expression levels" could be rephrased for better clarity.
Reviewer 2 Report
Comments and Suggestions for Authors
I think the authors have done a great job in the manuscript.
Figure 2: the tertiary structure shown for SfHsfB5 similar or different to other HsfB5 proteins? I think this comparison should be shown and mentioned.
How sure are the authors about the phenotypic changes observed are driven only by heat stress? In natural environment multiple stressors act together.
The authors use 45C as heat stress but what will happen when temp is freezing or too high? Any speculations?
There should be mention of how the authors plan to use this research in future. Are there any transgenic plants the authors are interested in making that can withstand high temp.
Comments on the Quality of English LanguageN/A
